# Psychological, sociocultural and economic coping strategies of mothers or female caregivers of children with a disability in Belu district, Indonesia

**Gregorius Abanit Asa**[1], **Nelsensius Klau Fauk**[2,3]*, **Paul Russell Ward**[2], **Karen Hawke**[4], **Rik Crutzen**[5], **Lillian Mwanri**[2]

1 Sanggar Belajar Alternatif (SALT), Atambua, Nusa Tenggara Timur, Indonesia, 2 College of Medicine and Public Health, Flinders University, Adelaide, South Australia, Australia, 3 Institute of Resource Governance and Social Change, Kupang, Nusa Tenggara Timur, Indonesia, 4 Infectious Disease—Aboriginal Health, South Australian Health and Medical Research Institute, Adelaide, Australia, 5 Department of Health Promotion, Maastricht University/CAPHRI, Maastricht, The Netherlands

* nelsen_klau@yahoo.com, fauk0001@flinders.edu.au

**Data Availability Statement:** All relevant data are within the manuscript and its Supporting Information files.

## Abstract

Caring for children with a disability can cause a range of psychological and socioeconomic challenges for parents and caregivers, such as anxiety, depression, inability to find affordable and appropriate childcare, loss of income and expenses related to disability specific treatment. As part of a study exploring the impacts of childhood disability on mothers or female caregivers and families, and the copy strategies they used, this paper describes strategies employed by mothers or female caregivers to cope with challenges associated with childhood disability within their family in Belu district, Indonesia. A qualitative approach using one-on-one in-depth interviews was used to collect data from participants (n = 22) who were recruited using a combination of purposive and snowball sampling techniques. Interviews were recorded, transcribed verbatim and imported to NVivo 12 for analysis. A qualitative framework analysis was used to guide data analysis. The conceptual framework of coping strategies guided the conceptualisation and discussion of the findings. The findings showed that active psychological coping strategies, including cognitive or acceptance strategies, knowledge of both health condition and socio-academic related development of children with a disability, and family relationship and support, were used by the participants to cope with psychological challenges facing them. Self-reliance and religious/spiritual coping strategies were also utilised. Sociocultural strategies, such as social withdrawal or disengagement, professional support and culture-based support, were used by the participants to cope with social impacts, stigma, and discrimination associated with childhood disability. Participants also reported using financial strategies such as selling of family assets to cope with the economic challenges. The findings indicate the need for programs and interventions that address the needs of mothers and female caregivers and their families, to assist with effectively managing the significant challenges they face when caring for a child with a disability. Further studies are needed, with a larger number of participants

**Funding:** The author(s) received no specific funding for this work.

**Competing interests:** The authors have declared that no competing interests exist.

and the inclusion of fathers or male caregivers, in order to better understand the broader coping experience of childhood disability impacts within families.

## Introduction

Disability is a major global public health problem which needs addressing. The current 2020 World Bank report and the World Health Survey estimate approximately 780 million or 15.6% of the world's population aged 15 years and older experience disabilities, and of those, one-fifth or between 110 million and 190 million people experience significant disabilities [1, 2]. The same survey shows that more than 90 million (5.1%) children globally, are estimated to live with a disability, of whom, 0.7% experience 'severe disability' such as deaf-blindness, intellectual disability or mental retardation and multiple disabilities [2, 3]. In Indonesia, the 2017 national socioeconomic survey reported over 37.1 million people live with a disability, and of this total number, just under 10% or 3.2 million, are children [4].

Childhood disability has psychological, social and economic impacts not only on a child with a disability but also the whole family, particularly parents and caregivers, such as mothers or female caregivers participating in this study and globally [5–8]. To respond to the impacts or challenges resulting from taking care of children with a disability, parents or caregivers are reported to use various coping strategies which include active processes and behaviours to manage continuing stressful situations [9–13]. For example, acceptance which refers to the capability to accept and care for a child despite limitations [14], has been reported as a strategy used by parents or caregivers to cope with the negative impacts of childhood disability [12, 15–17]. Religious coping such as praying, seeing faith or spirituality as the source of comfort, peace, hope, and the belief that God is in control of their life, is another strategy used by parents and caregivers [18–20], while social supports from healthcare professionals and other parents in similar situations, and support from close family members, such as spouses and grandparents, have also been reported as effective strategies used by parents to cope with difficult situations in caring for children with a disability [19, 21–24]. Problem-focused coping where parents put effort into finding appropriate solutions to the challenges associated with their child's disability [9, 25], is another strategy used.

These strategies are all underpinned by positive perceptions and behaviours of parents and caregivers, but there are also maladaptive coping mechanisms. Social withdrawal and avoidance coping strategies, such as hiding concerns regarding their children's difficulties, and avoiding social gatherings due to fear of negative judgement by others toward their children's challenging behaviours, are emotion focused and maladaptive strategies [26–29]. The maladaptive strategies are often used for several reasons. Parents and caregivers may believe that they do not possess the capacity to change or improve the condition of their children, hold the perception that they are not responsible for everything experienced by their children, be in denial of the diagnosis, and worry about or attempt to avoid social stigma and discrimination toward them and their children [25, 29, 30].

Although previous studies have investigated a wide range of coping strategies used by parents and caregivers of children with a disability, evidence around self-reliance, sociocultural and economic strategies is scarce [12, 25, 29–31]. In many developing countries, including Indonesia, mothers have the responsibility within the family to take care of children, husbands other household chores [32, 33] and are also vulnerable to negative impacts of childhood disability. Belu is reported as one of the districts in East Nusa Tenggara province with the highest

number of children living with a disability [34]. As far as is our knowledge, there is no information that provides evidence about strategies used by parents and caregivers to cope with the wide range of negative impacts of childhood disability in Belu, other districts in the province, or in Indonesia. This paper aims to fill this gap and focuses on exploring strategies used by mothers or female caregivers to cope with manifold burdens associated with caring for children with a disability in Belu district, Indonesia. The current findings contribute to the body of knowledge in public health domain and provide understanding on new childhood disability-related coping strategies which are still lacking in the existing literature [12, 25, 29–31]. The findings are presented separately from the impacts of childhood disability on mothers or female caregivers [8] as these are specifically about coping strategies used by mothers or female caregivers and need to be underpinned by existing literature on coping strategies. Understanding coping strategies of mothers or female caregivers is important to inform policy making and program development to address their needs and the needs of their families.

## Methods

The report of the methods section of this study was guided by the consolidated criteria for reporting qualitative studies (COREQ) checklist which contains 32 required items that support transparency and comprehensive reporting of qualitative studies especially interviews and focus groups (S1 Table) [35].

### Theoretical framework

The conceptual framework of coping strategies by Burr and Klein [36] was used in the current study. This framework suggests seven aspects of coping strategies; cognitive or acceptance, emotional, relationship, communication, community, spiritual, and individual development coping [12, 36]. Cognitive coping relates to the acceptance of a situation and other burdens experienced, acquiring useful knowledge and gaining new or different perspectives of the situation. Emotional coping refers to families expressing and sharing feelings and affection, resolving negative feelings and being more sensitive to other family members' emotional needs. Relationship coping refers to whether the situation experienced increases togetherness, adaptability, cooperation and tolerance among family members. Communication coping refers to members of the family being open and honest, understanding, listening to each other and sensitive to non-verbal communication. Community coping relates to whether a family could easily acquire support from others and whether the family meets the community's expectation. Spiritual coping means that a certain situation leads a family to be more actively involved in religious activities and at the same time increases their faith in God. Individual development coping refers to independence and self-autonomy. It also means that members of the family keep active in hobbies or things they like to do.

### Study setting

Belu district, the setting of the study, is located in the eastern part of Indonesia and shares the boarder with East Timor [8, 37]. It covers the area of 1,284.97 km2, population of 204,541 people including 100,922 male and 103,619 female [37, 38]. The total number of children registered with a disability in Belu is 348, of whom 58 children reside in two rehabilitation centres called *Pusat Rehabilitasi Hidup Baru* and *Bhakti Luhur* [34]. The remaining 290 children reside with parents or caregivers. Of the 22 participants in this study, 15 participants stay with their children with a disability and 7 participants' children reside in the rehabilitation centres. Of the 348 children, only 71 enrolled to a state special school for children with a disability

(*Sekolah Luar Biasa Negeri*, also known as SLBN) located in the district: elementary school (44 students), junior high school (14 students) and senior high school (13 students) level [8, 34].

## Study design and data collection

A qualitative inquiry was employed in this study and conducted from August to September 2019 in Belu district, Indonesia. The qualitative design is useful in understanding mothers or female caregivers' perceptions and experiences regarding how they have coped with challenges associated with caring for children with a disability within their family [39–41].

In-depth interview method was used to collect the data from a total of 22 participants. The interviews with each participant was conducted at a participant-researcher mutually agreed upon time and place, which was a private room at the special school for children with a disability or rehabilitation centres [8]. Participants recruitment processes employed a combination of purposive and snowball sampling techniques, starting with enlisting the help of the Principal of the state special school and the leaders of the two rehabilitation centres to distribute the study information sheet containing the contact detail of the field researcher (GAA) to potential participants by posting it on the information board at the school and rehabilitation centres [8]. This was followed by the snowball sampling technique where the initial participants who called to confirm their participation and were interviewed, were also asked to distribute the study information sheet to their friends, colleagues or other potential participants who might be willing to participate in this study. To be included in the study, participants had to meet the following criteria: (i) had to be a mother or female caregiver of a child with a disability, and (ii) had to be 18 years old or older. The interviews focused on exploring several key areas including: Participants' strategy to cope with childhood disability-associated challenges or impacts, availability and types of support from family members, friends and community, participants' perceptions of the condition of the children and corresponding challenges and strategies used to communicate any challenges to family or community members. The recruitment of participants ceased once data saturation had been reached as information or responses provided by a few last participants were similar to those of previous participants. Interview with each participant was one-on-one and face-to-face between the researcher (GAA/male) and the participant, with no other persons present in the interview room and took a duration of between 30 to 45 minutes. No participants withdrawal and no researcher-participant established relationship occurred prior to the interviews. A digital tape recorder was used to record the interviews and notes were also taken during the interviews. None of the participants took the opportunity to read and correct the transcription of the recorded information once it was offered to them after the interviews. The interviews were carried out in Bahasa [8].

## Data analysis

Before the analysis was undertaken, the audio recorded data were transcribed and translated into English by the first two authors (GAA and NKF). During this process data were cross checked and compared to maintain the quality and validity of the transcription and translation. The transcription and translation were then checked for accuracy and meaning by other authors (PRW, KH, RK and LM). The comprehensive analysis of the data was conducted using NVivo 12 software. Data analysis was guided by the five steps of qualitative data analysis in the of Ritchie and Spencer's framework analysis [42]. The five steps were: (i) *familiarisation* with the transcripts through reading data repeatedly, marking ideas, and giving comments to search for meanings, patterns, and ideas; (ii) *identifying a thematic framework* by making judgement and writing down key issues and concepts from participants; (iii) *indexing all the data* by making list of codes to look for similar or redundant codes and reduce them into

smaller number as codes referring to the same theme were grouped together to reach a few overarching themes and sub-themes. The themes were both derived from the conceptual framework for coping strategies and generated from the data; (iv) *creating a chart* through arrangement thematic framework so that data could be compared within each interview and across the interviews; and (v) *mapping and interpretation* data as a whole [8, 39, 42, 43]. This approach provides systematic structure to the management of qualitative data [42].

## Ethical consideration

Ethical approval of this research was received from Health Research Ethics Committee, Duta Wacana Christian University, Indonesia (No. 618/C.16/FK/2018). Prior to the interview, all recruited participants were informed about the objective of the research and that their participation was voluntary and that there would be no consequences if they chose to withdraw their participation during the interview. Participants were also informed that data or information they provided during the interviews would be treated confidentially and the transcripts would be anonymised. Prior to the interviews, participants were also informed about the interview duration (ca. 30–45 minutes), each interview being recorded using tape recorder, and notes being taken by the interviewer during the interview. Each participant was assigned a study identification letter and number (e.g. R1, R2) for de-identification purposes. Each participant signed and returned a written consent form on the interview day.

## Results

### Profile of participants

The age of the 22 participants included in the study ranged from 35 to 60 years, and were either a biological mother or female caregiver of a child with a disability. Of the 22 participating women, 14 graduated from junior high school, six graduated from senior high school, and two finished an elementary school. Most women (n = 18) reported to be unemployed and two respectively were household assistants and shopkeepers. Most of the participants were housewives from low socioeconomic family backgrounds (n = 18), as husbands were generally low-income workers such as carpenters, street vendors, vegetable and chicken merchants or farmers, and four women were unmarried or widowed. The types of disability affecting the participants' children are presented in Table 1. Participants reported various strategies used to cope with the impacts of childhood disability, including psychological, self-reliance, religion or spiritual, sociocultural, and economic coping strategies as explained below.

### Psychological coping strategies

**Cognitive or acceptance.**   Cognitive or acceptance strategy was used by mothers or female caregivers to cope. A few mothers reported feeling very shocked and stressed upon being

**Table 1. Characteristics of the children.**

| Disability | Children | Age range |
|---|---|---|
| | (n = 22) | (years) |
| Visual impairment | 3 | 8–13 |
| Hearing impairment | 3 | 9–15 |
| Speech impairment | 4 | 6–14 |
| Cognitive impairment | 3 | 10–15 |
| Mental & physical impairments | 9 | 6–16 |

informed by a health professional that their child had a disability, but over time and with a lot of thoughts and reflection, they accepted the diagnosis, relieving them from the worry and concerns:

> *"The first time my husband and I knew that he (her son) is autistic, it was like a nightmare. My husband and I were stressed out about everything. It took us about 3 months of endless thinking and reflecting to realise that we had to accept this" (R1: 38 years old mother).*

> *"I felt stressed out at that time (upon receiving a diagnosis). It took me about two weeks to gradually accept my child's condition. It was because at the beginning I was trying to figure out: was it my fault or my husbands' fault? . . . But then my husband and I changed our attitudes. He is our child. No parents want their children to live with a disability. So, even though we were stressed out with his condition but step by step we think positively and fully accept everything" (R19: 50 years old mother).*

Because they had noted abnormalities their children had, some mothers reported to have prepared to accept any condition their child might have prior to consulting healthcare professionals for confirmation. The preparation seemed to have helped parents overcome negative feelings upon receiving the untoward news of their children's diagnosis:

> *"I had already anticipated this before my husband and I talked to the doctor. My mother and I had seen some abnormalities in my son that is why we consulted with the doctor. After receiving the diagnosis my husband and I had a discussion about it. After the diagnosis, I was sad and stressed but I could handle it because I had prepared for it. Those feelings do not fully go away, sometimes I still feel sad but I can deal with it" (R14: 49 years old mother).*

> *"When my child was born, the nurse was reluctant to talk about my baby. She (the nurse) explained to me after I asked her for the second time. I said 'it is okay. I do not want to deny. I am ready for any condition of my child. . . Whether she is normal (non-disabled) or abnormal (disabled), she is my baby, my child. I will look after her. . . . . Even though I sometimes feel sad imagining her with normal (non-disabled physical condition) but I fully accept my daughter and her condition and deal with it because I have prepared and convinced myself to accept her condition before the nurse told me" (R7: 46 years old mother).*

**Knowledge of the health condition and socio-academic development of children with a disability.** Knowledge of the health condition and socio-academic related development of children with a disability seemed to be another strategy used by a few participants to cope with psychological impacts, such as feeling worried and afraid of the possibility of negative consequences that might happen to their children with a disability. Such knowledge was acquired through continuous consultation with healthcare professionals and teachers, and described by a few participants to help them overcome those feelings as they were certain about the health condition and development of their child:

> *"My daughter studies at special school for disabled children here (in Belu district). I always talk to the teachers and the school principal to check on my child academic development and progress, and her social relations with other kids at school. So, I know for sure my child is doing well. Knowing her progress and improvement makes me feel better and washes away negative feelings, like being worried too much" (R13: 47 years old mother).*

> *"I try to ask very detailed questions about my child's health condition every time we (the mother and her child) meet the doctor. I consulted the doctor about the kind of medicine or*

*supplement I need to give to my child. I want to make sure that my child is healthy. So, I am not afraid or worried about my child's health condition because I know he is fine" (R15:45 years old mother).*

The participants also described that initially they often felt worried and scared about the health condition and socio-academic development of their child due to the lack of knowledge about those aspects. This led to the initiative to talk with doctors, nurses, and teachers about their child's condition, which resulted in the relief of negative feelings as described below:

*"Initially, I had very limited knowledge about what is going on with my child. So, every time I noticed something different to him, I started to feel worried and scared because I did not know what happened and was just guessing. Then I started to talk to doctors about his health condition and got lots of information. This is a relief because I always refer to what the doctor said if I noticed something different to my child" (R17: 48 years mother).*

*"Before I actively talk to my child's class teacher, I was always worried about what she (her child) is doing at school, whether she is doing well or not, she listens to her teacher or not and how she interacts with her friends. But now when I have a chance, I encourage myself to ask her class teacher to know about all of these. It always feels good once the teacher says that she is doing well" (R3: 36 years old mother).*

**Family relationship and support.**   Parents with children with a disability drew support from family members, such as spouses, other children or grandmothers, as one of the coping strategies to deal with challenges. Talking directly to the child with a disability about their condition, playing, eating and praying together with them, dropping and picking them from school, and generating money to support family needs, were some instances of support from, or shared responsibility within the participants' family:

*"My husband is always concerned about our daughter. For example, when my husband is away for work, he always asks about our daughter condition or asks whether our daughter has eaten or not. This makes me feel better and calm because I know that my husband shares the responsibility and cares for our child" (R8: 53 years old mother).*

*"Her siblings like to play with her. They play inside or outside the house together. Once I asked my oldest son, are you embarrassed having a sibling with a disability? He said no. She is my sister. I continue supporting them to play together, share toys, get close to each other and understand each other. I feel good seeing them getting along very well. It makes me sometimes forget what my child (with a disability) has gone through." (R12: 52 years old mother).*

*"In the house there are only myself, my mother (the child's grandmother), my daughter (one with a disability) and my son. We do not know where the children's father is. My mother and I always work together. Sometimes, she drops my children off to school and then picks them up, while I work in a Chinese home as a housekeeper. If we do not have money, we discuss what stuff needs to be sold. It is hard, but I get support from my mother, so I do not get stressed out with the overloaded things to do" (R5: 45 years old mother).*

## Self-reliance or independence

Self-reliance or self-support strategy was used by a few participants to cope with childhood disability-related challenges. This was reflected in the decision to rely on their own capabilities in

handling daily tasks or taking care of children with a disability and the decision not to ask for support from relatives or extended family members. Though this strategy seemed to work effectively, participants also reported this coping method was challenging and stressful:

*"It is very challenging to take care of children with special needs like my child, but I have to deal with it. I do everything by myself: taking care of my children, feeding them, and doing the housework. I do not rely on my families (relatives or extended family members) because they might be busy with their life" (R2: 45 Years old mother).*

*"I have to do everything by myself with what I have. It is not easy, but I can take (care) of my child. My child is my responsibility. I have extended families living in this town. Some live in the village and some live near us but I do not ask for their help. I believe I can raise my child (with a disability)" (R11: 46 years old mother).*

Paradoxically, there was also an overall reluctance or unwillingness to seek help from extended family, neighbours and government. These women assumed people would judge them, and wanted to avoid feeling embarrassed, or enduring the possibility of being regarded as beggars or lazy persons by family members or other people within the community where they lived. Though self-reliance seemed to make them more independent in order to cope with challenges, this was underpinned by a feeling of acquiescence among the participants:

*"Although I have extended family here, I can't expect anything from them. I don't want to ask them money or other help. . ... I do not want to be seen like a beggar. I am embarrassed.. . . I do not intend to blame family or others, but the main important thing is we* (the women and her husband) *rely on ourselves to get over every difficult situation facing us. It is extremely difficult to raise our child, I mean the one with special needs but we try to do our best for our child" (R3: 39 years old mother)*

*"I do not know whether there is any government aid here for children with a disability or family having children with a disability. I never ask about it. I do not want to be seen as a lazy person who just hopes for aid. None of the government officials tell me about it. I do not ask for help from the neighbours either, I do not want to be disdained by other people. So far, my husband and I rely on ourselves and handle everything without any support from others" (R14: 48 years old mother).*

## Religious/spiritual coping strategy

All participants reported that praying to God was a spiritual resource and used to cope with the difficulties they experienced. Religious practices such as praying at home, going to church, participating in spiritual community activities, were described to make them feel calm and peaceful. They described having faith in and relying on God and ancestors, with the hope that they (the ancestors and God) would break down barriers and thus show them the best course to take:

*"I pray to God every day to help me out of the difficulties I face. I also always go to the church and participate in Legion Maria organisation (catholic religion) where I can pray for myself and others. I pray to my ancestors as well. I do believe that they watch me and listen to my prayers. I feel calm and less stressed every time I visit the church and pray to God and my ancestors" (R4: 40 years old mother)*

*"I often think that there might be something wrong in the past and now I have to repay debts for my previous life experience. But I do not know what that was. I light up candles for my*

*ancestors in their cemeteries and for God, and hope that my ancestors and God would show the right way to overcome all difficulties facing me. Letting my ancestors and God know about my situation and what my family is going through makes me feel peaceful" (R2: 42 years old mother).*

*"Several prayer teams (tim doa) have visited us and prayed for my daughter. I feel calm. They encouraged me to have strong faith in God" (R11: 46 years old mother).*

*"I can only pray to God and hope that God will help my granddaughter in His own way" (R12: 59 years grandmother caregiver).*

## Sociocultural coping strategies

**Social withdrawal and professional support.** Social withdrawal or disengagement was used by the participants as a strategy to cope with social discriminatory and stigmatising attitudes and behaviours, which often led to the women feeling hurt, sad and angry. Several participants preferred not to actively participate in social activities and extended family gatherings, in the hope that they would avoid negative attitudes and behaviours of others towards them and their child with a disability. The decision for social disengagement seemed to be based on the experience of past disability-related discriminatory and stigmatising attitudes and behaviours:

*"He (my son with disability) often laughs out loud by himself. One day when I took him to the community activity, other children around him laughed at him because he laughed by himself. It hurts. So, I have to think twice whether I want to participate in the community activities or not (if she wants to participate then she must take her son with her because nobody else takes care of him). Often, I decide it is better to not participate or I give an excuse by saying I am busy taking care of my son. My child might not understand the way other people react to him, but I do and can feel how people look at him. I do not want people to hurt us through their negative attitudes and behaviours towards my child. So, it is better for us not to involve in social activities. Other people and kids do not understand our child's condition and the challenges facing our family" (R8: 39 years old mother).*

*"I hardly attend community prayer because I do not want other children laughing at my son. It happened that children around him burst into laughing when he could not finish the Mother Marry pray (Catholic pray). It made me feel so sad. Since that time, we (the mother and her son) choose not to participate in social activities within our community. It hurts me a lot if other people or children do negative things to him. I think it is better to be like this (social disengagement) to avoid those negative reactions from others" (R6: 49 years old mother).*

Seeking support from healthcare professionals was used by the participants not only to cope with stigma and discrimination associated with childhood disability, but also to educate others in order to create a positive and supportive environment around their child. Some of the women reported disability-related stigma experiences to healthcare professionals. They asked health professionals to provide information and raise awareness in the community about the impact of stigma towards a child with a disability:

*"Negative treatment of other kids towards my child bothers me a lot. It does not feel good when you see your child is treated in a certain way you do not like. I talked to the doctors and nurses about this experience and asked them to do awareness raising activities or give some information to community members during the monthly visit (posyandu) so that adults and*

*children do not stigmatise the ones with special needs like my child. I know that they (health-care professionals) have done it (awareness raising) twice. At least I now know other parents are aware that there have been negative treatments towards children with a disability. I hope parents talk to their children about the impact of negative treatment or advise other people to not do that"* (R17: 43 years old mother).

*"I told the midwife at the integrated healthcare centre about the negative experiences we have had due to my child's condition. At the end of the service the midwife provided some informa-tion and asked the other parents (mothers) who attended that activity to treat children with a disability kindly and with love, and help them to grow within our community. I was delighted that the midwife directly responded to the problem and I am sure other women have also talked to their children or husbands about this"* (R3: 46 years old mother).

*"I told a nurse, who is a friend of mine, about how other children have treated or reacted to my nephew's condition and she said they (healthcare professional) will do something about it, but I haven't heard back from her"* (R2: 48 years old female caregiver/aunty).

**Culture-based support.**    Seeking support from other people who shared the same cultural values and perspectives about children with a disability was another strategy used by women to cope with the disability related stigma and discrimination. Women drew comfort and strength from culture-based perceptions that every human being is an image of God. As such children with a disability are perceived as being lucky, and a good fortune for a family. Women felt that these beliefs were the reasons for the positive attitudes and behaviours towards children with disability within their culture. A few women described that they some-times shared their stories of challenging experiences associated with childhood disability to other mothers with the same cultural background, and felt encouraged and supported:

*"In our culture, a child with a disability is not a disgrace for a family. Thus, we do not allow negative attitudes or behaviours towards children with a disability. For us, children are for-tune to family. I am aware that sometimes other people who do not hold the same view as us, have condescending look at and stigmatising attitudes towards my child but I am sure that many other people (who share the same culture as her) are supportive. As a mother, I often share the burden I experience with a few close friends of mine (other mothers with the same cultural background) and they encourage me a lot"* (R22: 45 years old mother).

*"I was told by my parents and grandparents to look at everybody with respect regardless of their social status and physical appearance because everybody is the image of God. This is what we inherit in our culture. A child with a disability is a fortune for family, not a disgrace. I often talk to a friend of mine (a mother with the same cultural background as her) if I am pissed off with discriminatory behaviours of other kids and also some grownups towards my child. She encourages me a lot, supports me emotionally and advises me to ignore the negative words I heard about my child"* (R20: 39 years old mother).

### Economic coping strategies

**Selling family assets, pawning wedding jewellery, and borrowing money.**    The sale of family assets seemed to be a common strategy used by the participants to cope with the increased healthcare expenditure and other needs of their child with a disability and other family members. Several participants described to have sold their family assets such as motorbike and land to cope with the economic hardships facing their family, as illustrated in the following assertations:

*"I sold his father's motorbike because no one can ride it since his father run away (left them). All the money I got from selling the motorbike was used for the treatment of my son, our daily needs and his transportation cost to school every day"* (R1: 46 years old mother).

*"We (the mother and her husband) had four pieces of land. But we have sold two of them to cater family and the needs of our child with a disability. We sold them cheaper than the market price because we were really in need"* (R6: 48 years old mother)

Several women described pawning wedding jewellery and borrowing money from neighbours as strategies to cope with the family needs and the needs of their child with a disability. These strategies were described to be helpful especially at the time when they were in need for basic necessity for family and had no other alternatives of support at all:

*"To be honest I used to pawn my wedding ring and necklace in a pawnshop. I did that because at the time I really needed money to buy foods for family and fulfil the needs of my child (with a disability) but had no other choices. I always told my husband after I pawned my ring or necklace, so he knew that we had to get back our wedding jewellery by paying the loans and interest"* (R3: 47 years old mother).

*"I think some of my neighbours here know well my economic situation. They know what my husband and I do for living. My husband is a carpenter and I work in a Chinese house. I often borrow money from some neighbours with no interest to fulfil the needs of my little one who has special needs. It is helpful because there is no time limit to return but you know, the problem is if we do not have enough money for long period of time, we are embarrassed to pass their home (neighbour) or embarrassed to meet them or just to say hello. They (neighbours) might remember the loan given"* (R9: 46 years old mother).

**School absenteeism.**    Deliberate school absenteeism of children with a disability was also reported as an economic coping strategy. Several participants described that sometimes they could not send their child with a disability to school due to lack of finances, which impacted them being able to afford transportation. School absenteeism of children with a disability was a strategy undertaken to reduce the expenses and reallocate the budget for the basic needs of the family, medicine, and school fees of other children:

*"The motorbike taxi (ojek) cost for my kid is IDR 4,000, meaning IDR 8,000 every day (USD 1 = ±IDR 15,000). Sometimes, if we did not have money to pay the ojek then I asked him (her child with a disability) not to go to school. Sometimes, I decided not to send him to school to reduce the expense and use the money to buy foods for family"* (R1: 46 years old mother).

*"He (her child with disability) often asked me why he did not go to school. I told him that today we (the women and the child's father) did not have money to pay the ojek because we had spent the money on medicine and his brother's school fee"* (R18: 48 years old mother).

## Discussion

This paper explores strategies used by mothers or female caregivers of children with a disability in Belu, Indonesia to cope with a range of childhood disability-related challenges. Consistent with the constructs of coping strategy framework used in this study and the findings of previous studies [12, 15–17, 36, 44], the current study suggests that learning to accept (cognitive or

acceptance coping) the condition of their children with a disability was used by these women as a proactive strategy to cope with psychological challenges. In addition to the existing knowledge about acceptance coping, it suggests that being prepared at the time of diagnosis, such as talking with healthcare professionals, helped the participants to accept their child's condition and cope with negative psychological consequences, such as feeling sad and stressed following the diagnosis. These are in line with the findings of a previous study employing the same coping strategy framework, which reported information and acceptance as the most frequent coping strategies used by parents [12].

Our findings demonstrate childhood disability-related coping strategies which have not been reported in previous studies [25, 30]. For example, taking initiative to procure knowledge of the disability in order to understand it and to help others understand it to best support the child helped mothers or female caregivers to cope with psychological consequences related to childhood disability. Similarly, understanding socio-academic-related development of children with a disability, such as their social relations with other children with a disability at school and their academic performance, was additional strategy used by women to cope with negative psychological consequences facing them. The knowledge coping strategy was developed by the participants through sustained consultations with doctors and nurses who provided medical services for their children and with the teachers at special school where their children were enrolled. Knowledge of the health condition and socio-academic development of their child was a proactive or problem-focused strategy, which assisted them to feel calm and reduced negative feelings, such as being afraid and worried about the children's condition. These findings support previous studies [12, 31] where mothers' knowledge of childhood disability-related laws, available resources, services and supporting system, and process/structure within the school system have been reported to help women cope with the situation related to their child's condition, resulting in stress relief. Similarly, supporting the findings of previous studies [19, 21–24] and the constructs of coping strategy framework [36], this study suggests that family support or shared responsibility among close family members, including spouses, children and mothers, are a healthy problem-focused strategy that most of the participants used to cope with disability-related challenges. These supportive environments led to psychological benefits, such as feeling better, calmer, and not being stressed out when caring for their children with a disability. Seeking support from healthcare professionals, such as nurses and midwives to prevent social stigma and discrimination towards children with a disability and their families was also an effective problem-focused coping strategy. In addition to sharing stories with healthcare professionals about their own experiences of stigmatising and discriminatory attitudes and behaviours, participants asked healthcare professionals to deliver awareness raising activities or information sessions about disability-related stigma prevention to the community members where the participants lived. The act of asking the healthcare professionals to advocate for the patients, aligns with known health practices such as those described in the Australian Charter of Healthcare Rights where the rights of consumers, or someone they care for, can be expected when receiving health care [45]. While this an Australian health care charter, consistent with principles of the Ottawa Charter for Health Promotion [46], designing a health care that brings consumers, clinicians, healthcare managers and policy-makers together produces a programs that can be cost-effective and relevant to all involved.

The use of culture-based support from friends or other mothers with the same cultural background was described by some participants as a strategy that was encouraging and supportive, not only to help them cope but to also view the disability in a positive way than being a negative and shameful thing. These cultural beliefs made the women feel empowered and grateful for their children who lived with a disability. Likewise, religious or spiritual coping strategy through activities, such as praying to God or ancestors, going to church, participating

in spiritual community activities, which has been used by many other parents of children with a disability in other settings globally [18–20, 36], was also a beneficial coping strategy. The women described feeling calmer, more peaceful, and less stressed about the disability and challenges surrounding it. The use of these strategies was due to the participants' strong beliefs and expectations that God and ancestors watch and listen to them and show solutions to problems facing them.

Additionally, in supporting previous studies [27, 28], the current findings suggest the use of a potentially maladaptive coping mechanisms such as social withdrawal or disengagement strategies to avoid discriminatory and stigmatising attitudes and behaviours. Disengagement from social or community activities was described by several participants, due to the perception that community members judged their family and did not understand their child's condition and challenges facing them. These women felt that social engagement might lead to their child and family being hurt. The justification to use these maladaptive strategies seemed to be based on the perceptions that the behaviours of a child with a disability do not conform with societal expectations. Such perceptions, as reported previously [29], can lead to the belief among parents that children living with a disability are not welcomed in certain environments, which leads to the reduction of interactions with friends and various social groups leading to social isolation, which is one of the known determinants to poor health [47]. Previous studies [29, 48, 49] have reported social withdrawal or disengagement coping strategy as being maladaptive as it can lead to missed opportunities to receive social supports and engage in healthy social activities, which are important when much of a parent's time is consumed with thinking about or caring for a child with a disability. Though the capacity for self-reliance was commendable in this cohort of women, it would seem from the perceptions and beliefs as outlined above, that these decisions were borne out of acquiescence, where the women felt a resigned acceptance to cope on their own, rather than having a true desire to. They described feeling reluctant to solicit support due to feeling embarrassed, and there were reports of active avoidance to seek help, due to fear of negative perceptions from others, such as being regarded as beggars and lazy people, so these women reported isolating themselves and their family from others, which is concerning.

Some women reported financial stress due to caring for a child with a disability. Selling assets, borrowing money, and pawning wedding jewellery, were some of the financial strategies employed to manage economic hardship. The underlying reason for these was the unavailability of other choices, families were in urgent need of money for medical and transport costs associated with their disabled child, and to provide for other family members. Literature shows [50, 51] that many families in low- and middle- income countries, have reported borrowing money and selling family assets as common strategies to cope with family healthcare expenditures. School absenteeism of children with a disability due to unaffordability of transportation costs, and reallocation of the family budget for basic needs, were also strategies used by the women we interviewed. These economic coping strategies have been reported by the heads of families caring for AIDS-orphaned children that the orphaned children were withdrawn from school to engage in income generating activities to support the needs of the family [43, 52, 53].

## Limitations and strengths of the study

There are several limitations that need consideration in interpreting the findings. Firstly, the study was restricted to mothers or female caregivers in Belu, thus the study reflects the unique coping experiences of females in this district, which may be different to mothers or female caregivers in other settings especially in developed countries with available well-structured

support system for people with a disability. Secondly, we explored perspectives of mothers and female caregivers and while a great deal of information was gleaned from these interviews, it is an incomplete holistic overview of coping strategies used by families, including male caregivers, that face many obstacles when caring for children with a disability. It was a conscious choice to start with mothers or female caregivers as participants because like in many other developing countries, women in Indonesia and Belu in particular have the responsibility within the family to take care of children, husbands other household chores [32, 33] and are vulnerable to negative impacts of childhood disability. The women interviewed were also at different stages in terms of time since disability diagnosis, so only conducting one interview gives us a very good snapshot, but not an indication of their coping mechanisms over time. However, our findings represent an initial qualitative exploration of coping strategies used by mothers and female caregivers of children with a disability, which is very useful to inform disability-related programs and interventions that address impacts of childhood disability on parents, caregivers and families in Belu and other similar settings in Indonesia and globally.

## Conclusion

The study findings indicate that mothers and female caregivers use a variety of coping mechanisms to cope effectively with the impacts of childhood disability based on their living situations, culture, religion, family support and available economic resources. Some of the participants use more positive, proactive and problem-focused coping mechanisms and strategies, such as acceptance of their child's condition, procuring knowledge of the health condition, and being aware of the socio-academic-related development of children with a disability. Utilisation of family support, social support from friends with similar cultural backgrounds, and support from healthcare professionals, are also healthy strategies employed, which seem to enable them to cope with the significant psychological and social challenges surrounding them and their family. Some women apply more passive coping strategies, such as self-reliance and religious coping, and others use maladaptive coping strategies such as social withdrawal, avoidance, and disengagement, which are not conducive to creating and maintaining a good support system for the women themselves, but also for their family, including the child with a disability. These negative ways of coping can also lead to the loss of opportunities around social support and engagement in activities outside of parenting, such as hobbies and sports. Economic strategies such as the selling of family assets, pawning wedding jewellery, borrowing money and withdrawing disabled children from school, while understandable, are only temporarily helpful, and can have negative impacts long term. It is crucial that there is consistent support, and increased education and awareness around child disability and associated stigma and challenges. It is crucial to the wellbeing of disabled children, to understand why different families employ different strategies, including fathers and male caregivers, so tailored programs and interventions can be created, that best support families' psychological, physical, emotional, mental and economic wellbeing. Our findings provide insight into coping strategies of mothers and females caring for a child with a disability, but further research is needed to provide a more holistic picture of the challenges, limitations and needs parents of children disability. This research should include perspectives of parents, other family members and caregivers, community and healthcare professionals.

## Supporting information

**S1 Table. Consolidated criteria for reporting qualitative studies (COREQ): 32-item checklist.**
(DOCX)

## Author Contributions

**Conceptualization:** Gregorius Abanit Asa, Nelsensius Klau Fauk.

**Formal analysis:** Gregorius Abanit Asa, Nelsensius Klau Fauk.

**Investigation:** Gregorius Abanit Asa.

**Methodology:** Gregorius Abanit Asa, Nelsensius Klau Fauk, Paul Russell Ward, Karen Hawke, Rik Crutzen, Lillian Mwanri.

**Project administration:** Gregorius Abanit Asa, Nelsensius Klau Fauk.

**Writing – original draft:** Gregorius Abanit Asa, Nelsensius Klau Fauk.

**Writing – review & editing:** Gregorius Abanit Asa, Nelsensius Klau Fauk, Paul Russell Ward, Karen Hawke, Rik Crutzen, Lillian Mwanri.

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
