## [Decision Letter · Decision Letter 0]

9 Feb 2021

PONE-D-20-32757

Psychological, sociocultural and economic coping strategies of mothers or female caregivers of children with a disability in Belu District, Indonesia

PLOS ONE

Dear Dr. Fauk,

Thank you for submitting your manuscript to PLOS ONE. After careful consideration, we feel that it has merit but does not fully meet PLOS ONE’s publication criteria as it currently stands. Therefore, we invite you to submit a revised version of the manuscript that addresses the points raised during the review process.

We look forward to receiving your revised manuscript.

Kind regards,

Joseph Telfair, DrPH, MSW, MPH

Academic Editor

PLOS ONE

Additional Editor Comments:

Due to the challenges of finding qualified reviewers for this submission, the editor served as the alternate. The authors re advised to attend to the comments from the reviewer to considered for moving to the next stage.

Journal Requirements:

Reviewers' comments:

Reviewer's Responses to Questions

**Comments to the Author**

1. Is the manuscript technically sound, and do the data support the conclusions?

Reviewer #1: Yes

2. Has the statistical analysis been performed appropriately and rigorously? 

Reviewer #1: N/A

3. Have the authors made all data underlying the findings in their manuscript fully available?

Reviewer #1: Yes

4. Is the manuscript presented in an intelligible fashion and written in standard English?

Reviewer #1: Yes

5. Review Comments to the Author

Reviewer #1: Dear editor,

Thank you for the opportunity to review this manuscript. Below are my comments.

Title: Informative

Abstract: Informative.

Introduction: Appropriate and informative, contain information on impacts and coping strategies.

Methodology: Informative. However, authors should indicate how many participants whose children reside in the rehabilitation centres and how many parents who stay with their children?

Why was snowball used because the principal and the leaders of the 2 rehabilitation centres know the parents or caregivers of these children or have records of their contact details? Meaning they should have contacted them to distribute the information sheet rather than the researchers asking the initial participants.

Results: The results are informative, descriptive with details

Discussion: Appropriate and informative.

Conclusion: Appropriate and informative.

References: Many references (>60%) were published before 2016

6. PLOS authors have the option to publish the peer review history of their article (what does this mean?). If published, this will include your full peer review and any attached files.

Reviewer #1: No

---

## [Author Response · Author response to Decision Letter 0]

14 Feb 2021

Responses to reviewers' comments are attached.

---

## [Decision Letter · Decision Letter 1]

23 Apr 2021

Psychological, sociocultural and economic coping strategies of mothers or female caregivers of children with a disability in Belu District, Indonesia

PONE-D-20-32757R1

Dear Dr. Fauk,

We’re pleased to inform you that your manuscript has been judged scientifically suitable for publication and will be formally accepted for publication once it meets all outstanding technical requirements.

Kind regards,

Joseph Telfair, DrPH, MSW, MPH

Academic Editor

PLOS ONE

Additional Editor Comments (optional):

Reviewers' comments:

Reviewer's Responses to Questions

**Comments to the Author**

1. If the authors have adequately addressed your comments raised in a previous round of review and you feel that this manuscript is now acceptable for publication, you may indicate that here to bypass the “Comments to the Author” section, enter your conflict of interest statement in the “Confidential to Editor” section, and submit your "Accept" recommendation.

Reviewer #1: All comments have been addressed

2. Is the manuscript technically sound, and do the data support the conclusions?

Reviewer #1: Yes

3. Has the statistical analysis been performed appropriately and rigorously? 

Reviewer #1: N/A

4. Have the authors made all data underlying the findings in their manuscript fully available?

Reviewer #1: Yes

5. Is the manuscript presented in an intelligible fashion and written in standard English?

Reviewer #1: Yes

6. Review Comments to the Author

Reviewer #1: Dear authors. Thank you for addressing all the comments and aligning the manuscript according to the journal guidelines.

7. PLOS authors have the option to publish the peer review history of their article (what does this mean?). If published, this will include your full peer review and any attached files.

Reviewer #1: No

---

## [Editor Report · Acceptance letter]

27 Apr 2021

PONE-D-20-32757R1 

Psychological, sociocultural and economic coping strategies of mothers or female caregivers of children with a disability in Belu District, Indonesia 

Dear Dr. Fauk:

I'm pleased to inform you that your manuscript has been deemed suitable for publication in PLOS ONE. Congratulations! Your manuscript is now with our production department. 

Kind regards, 

on behalf of

Dr. Joseph Telfair 

Academic Editor

PLOS ONE